# Present Status and Future Trends of Natural-Derived Compounds Targeting T Helper (Th) 17 and Microsomal Prostaglandin E Synthase-1 (mPGES-1) as Alternative Therapies for Autoimmune and Inflammatory-Based Diseases

**DOI:** 10.3390/molecules25246016

**Published:** 2020-12-18

**Authors:** Anella Saviano, Federica Raucci, Gian Marco Casillo, Chiara Indolfi, Alessia Pernice, Carmen Foreste, Asif Jilani Iqbal, Nicola Mascolo, Francesco Maione

**Affiliations:** 1ImmunoPharmaLab, Department of Pharmacy, School of Medicine and Surgery, University of Naples Federico II, Via Domenico Montesano 49, 80131 Naples, Italy; nella.1993@hotmail.it (A.S.); federica.raucci@unina.it (F.R.); gianmarcocasillo@virgilio.it (G.M.C.); chiara_indolfi@hotmail.it (C.I.); alessiapernice.ap@gmail.com (A.P.); carmenforeste@gmail.com (C.F.); A.J.Iqbal@bham.ac.uk (A.J.I.); 2Institute of Cardiovascular Sciences (ICVS), College of Medical and Dental Sciences, University of Birmingham, Birmingham B15 2TT, UK

**Keywords:** BELFRIT, immunity, mPGES-1, natural compounds, Th17

## Abstract

Several natural-based compounds and products are reported to possess anti-inflammatory and immunomodulatory activity both in vitro and in vivo. The primary target for these activities is the inhibition of eicosanoid-generating enzymes, including phospholipase A2, cyclooxygenases (COXs), and lipoxygenases, leading to reduced prostanoids and leukotrienes. Other mechanisms include modulation of protein kinases and activation of transcriptases. However, only a limited number of studies and reviews highlight the potential modulation of the coupling enzymatic pathway COX-2/mPGES-1 and Th17/Treg circulating cells. Here, we provide a brief overview of natural products/compounds, currently included in the Italian list of botanicals and the BELFRIT, in different fields of interest such as inflammation and immunity. In this context, we focus our opinion on novel therapeutic targets such as COX-2/mPGES-1 coupling enzymes and Th17/Treg circulating repertoire. This paper is dedicated to the scientific career of Professor Nicola Mascolo for his profound dedication to the study of natural compounds.

## 1. Introduction

The Belgian decree on botanicals, published 20 years ago, was the prototype of legislation and regulatory practice used in several European countries. In 1997, Belgium was one of the first countries to introduce a notification procedure and scientific risk evaluation by an Advisory Commission. Contextually, the authorities of Belgium, France, and Italy, each assisted by renowned scientific experts, decided to develop a common approach to evaluating botanicals [1]. As a first step, the three parties drafted a standard list of traditionally used plants safe to use in food supplements, commonly known as the BELFRIT project or BELFRIT list [2,3]. This evolved with the support of the European Commission and Member States and now guarantees the safety, quality, and effectiveness to health promoting prosperities of food supplements. Most other Member States acknowledge the importance of European Union (EU) harmonization in this “self-growing” area [4,5,6].

Modern medicine makes use of many plant-derived products/compounds as the basis for pharmaceutical drugs [7]; and quite often it applies modern standards of effectiveness testing to herbs and medicines derived from natural sources, performs high-quality clinical trials and uses standards for purity or dosage [8]. In this scenario, many botanicals are used in both food supplements and nutraceuticals, and yet a precise, unique, and standardized definition/s and procedure/s are still missing.

In this opinion paper, we provide a current state of the art about the anti-inflammatory and immunomodulatory properties of natural-derived compounds (including nutraceuticals, functional food, and dietary supplements) targeting microsomal prostaglandin E synthase-1 (mPGES-1) and the T helper 17 cells (Th17) and regulatory T-cells (Treg) axis, in order to provide a scientific rationale for their potential therapeutic use. Further investigation, in both pre-clinical and clinical fields, are required to provide in-depth evaluation of these botanicals and their bioactive components in the context of autoimmune and inflammatory-based diseases for health-promoting and disease-preventing purposes.

Nutraceuticals, functional foods, and dietary supplements have been known to exert beneficial effects against a variety of disease conditions [9,10,11]. Several medicinal plants and their isolated components have also been identified to possess health-promoting properties [12]. Moreover, a varied diet containing certain phytochemicals or introduced through supplementation have shown potential antioxidant, anti-inflammatory, and immunomodulatory benefit [13,14]. Therefore, the role of natural products and food is essential in maintaining and/or improving immune function [15,16]. Functional foods are those enriched or enhanced that provide health benefits beyond essential nutrients when consumed at efficacious levels as part of a varied diet. These mainly include berries, fermented dairy products, green tea, garlic, citrus fruits, and other herbal formulations [17,18,19]. In contrast the term “dietary supplement” describes a broad and different category of products (mainly containing vitamin C, vitamin D, minerals, omega-3 fatty acid, docosahexaenoic acid, etc.) that we eat or drink to support good health and supplement the diet. While nutraceuticals are food components, such as polyphenols, flavonoids, carotenoids, saponins, sulfides, which are derived from food sources with extra health benefits in addition to fundamental nutritional value [20,21,22,23,24].

These products could work both at the cellular and molecular level by triggering immune cells, up-regulating immune-related genes, and manipulating the systemic immune system, thereby providing natural immunotherapeutic options. These cellular and molecular mechanisms of natural products are essential to define the possible molecular and cellular targets, which could pave the way for discovering novel natural products/compounds exerting the immune-boosting effects [25,26].

## 2. Selected Studies and Inclusion and Exclusion Criteria

All studies were selected through a Medline–PubMed search using different combination of terms or keywords such as Th17, COX-2, mPGES-1, medicinal plants, and natural compounds/products. We have identified only original articles in English that evaluated pre-clinical studies, in vivo rodents models, and isolated and/or well-characterized compounds/extracts from all articles. Studies that analyzed natural substances purchased commercially were not excluded in this work. The next selection was to consider reports in which the method adopted used a natural compound in a well-established disease. The final selection criteria was choosing medicinal plants according to their presence in the Italian list of botanicals and the BELFRIT list.

## 3. Natural Compounds Targeting COX-2/mPGES-1/PGE_2_ Cascade in Inflammatory-Based Diseases

Inflammation is a complex protective mechanism against noxious stimuli of chemical, physical, and/or biological origin, characterized by molecular and cellular defensive responses aimed at resolution of ongoing inflammation and restoration of tissue integrity [27,28,29]. However, the persistence of inflammatory inducers and the alteration of processes directed to homeostasis restoration can lead to the onset of chronic inflammation [30,31]. Inflammatory processes are generally associated with the innate immune system, but scientific evidence has shown that innate and adaptive immune cells collectively orchestrate the inflammatory response and that adaptive immune components are also involved in the production of memory cells that can sustain the chronic nature of inflammation-driven by the innate arm [32]. During the past decade, considerable progress has been made in understanding the cellular and molecular events in the acute inflammatory response and the role primary mediators have in infection and tissue injury [33]. Every early immunity “battle” begins with neutrophils, quickly recruited to sites of inflammation for an early response against the noxious stimulus under close control of several endogenous mediators [34]. This phlogistic scenario correlates with a transient increase of pro-inflammatory factors including (i) cytokines such as the interleukin (IL)-1 family (IL-1α/β), IL-6, and tumor necrosis factor-α (TNF-α) that are involved in the early stages of inflammatory and immune processes and warn the host to induce an inflammatory reaction against pathogens [35]; (ii) chemokines for the control of leukocyte extravasation and chemotaxis towards the affected tissues; (iii) the complement fragments (C3a, C4a, and C5a also known as anaphylatoxins) that promote granulocyte and monocytes recruitment and induce mast-cell degranulation [36]; and (iv) prostaglandins (PGs), in particular PGE_2_, one of the principal lipid mediators of the Arachidonic acid cascade [37]. The first sign of recovery is the switch from PGs (pro-inflammatory lipid mediators) to lipoxins and resolvins (anti-inflammatory lipid mediators); indeed, it inhibits the recruitment of neutrophils and promotes the recruitment of monocytes, restoring homeostatic conditions [38,39].

In the last few years, in the context of acute inflammation, the scientific community has focused the spotlight on mPGES-1 enzyme that acts as a crucial regulator in the terminal steps of PGE_2_ production from intermediate PGH_2_ [40]. The baseline expression of mPGES-1 in different tissues is low, but in response to inflammatory stimuli and cytokines such as lipopolysaccharide (LPS), IL-1β, TNF-α, and IL-17A, mPGES-1 is up-regulated and functionally coupled with COX-2 to mediate pro-inflammatory PGE_2_ production [41,42,43,44]. This elevation in lipid mediators is implicated in the pathogenesis of several inflammatory diseases such as gouty arthritis, rheumatoid arthritis (RA), and atherosclerosis [45,46,47].

Selective inhibition of downstream mPGES-1 [48] for a specific reduction in PGE_2_ production is proposed as a safer alternative compared with nonsteroidal anti-inflammatory drugs (NSAIDs) [49,50]. A variety of compounds which target mPGES-1 have been described in the literature and are summarized in Figure 1 [51,52,53,54]. Of particular interest is baicalin, a bioactive flavone extracted from the root of *Scutellaria baicalensis* Georgi. It has been reported that baicalin and, its aglycone baicalein not only suppress the overexpression of pro-inflammatory mediators such as nitric oxide (NO), PGE_2_, TNF-α, IL-1β, and IL-6 [55,56] but also inhibit the expression of inducible enzymes COX-2 and inducible nitric oxide synthase (iNOS) [57]. Similarly, the main component of turmeric *Curcuma longa* has been reported as an efficacious agent against both PGE_2_ production, COX-2 expression [58], and Matrix Metalloproteinase (MMPs) secretion [59].

Furthermore, polyphenols, such as naringenin and hesperetin from *Citrus aurantium* [60], ellagic acid from *Punica granatum* [61], apocynin from *Picrorhiza kurroa* [62,63], hyperoside from *Hypericum perfoaratum* [64], mangiferin from *Mangifera indica* [65], and alkaloids as berberine (from *Coptis japonica*) are all capable of attenuating the severity and extension of intestinal-inflammatory injuries via inhibition of neutrophil infiltration, pro-inflammatory proteins COX-2, iNOS, and nuclear factor kappa-light-chain-enhancer of activated B cells (NF-κB) activation [66]. Notably, mangiferin also presents analgesic properties due to its ability to reduce pain via PGE_2_ reduction [67]. 

A series of in vivo studies found that cedrol, from *Zingiber officinale* [68], leonurine from *Leonurus cardiaca* [69], and madecassoside, triterpenoid isolated from *Centella asiatica* [70], can effectively alleviate inflammatory response through the inhibition of inflammatory pathway COX-2/mPGES-1/5-Lipoxigenase (5-LO) and the up-regulation of anti-inflammatory molecule IL-10. 

Moreover, recent work examining the actions of glycyrrhizin (from *Glycyrrhiza glabra*) [71,72], cryptotanshinone and tanshinone IIA (from *Salvia miltiorrhiza* Bunge) [73], juglanin (a natural compound derived from the crude *Polygonum aviculare*) [74], and gingerol (from *Aframomum melegueta*) [75], in neuroinflammation and nociception, indicate that these natural compounds are potential therapeutic agents in neurodegenerative diseases and painful conditions.

Finally, the anti-inflammatory and analgesic properties of carnosol, carnosic acid (diterpenoids isolated from *Salvia Officinalis*) [76] and ginsenoside K (saponins from *Panax ginseng*) [77] have been intensively described for their ability to modulate pathways involved in inflammation and painful syndromes, including COX-2, mPGES-1 and 5-LO. 

These findings pave the way for the use of these botanical-derived compounds as novel anti-inflammatory and analgesic agents targeting COX-2/mPGES-1 pathway.

## 4. Natural Compounds Targeting Th17/Treg Axis in Immune-Mediated Inflammatory Diseases

Considering inflammation from a “cellular point of view”, although neutrophils and macrophages have traditionally been looked upon as dominant cell types during the resolution phase, accessory cells such as Th17 and Treg have more recently emerged as important players during resolution. They may link innate and adaptive immune systems [78]. CD4 T cells regulate several immune answers in order to fight against different disease-causing noxious stimuli. The binding of T cell receptor (TCR) to the peptide–major histocompatibility complex (MHC) activates naïve CD4 T cells that differentiate into effectors cells, including Th17 and Treg [79,80]. Th17, which express the transcription factor retinoic acid receptor-related orphan receptor γt (RORγt), arbitrate immune responses against extracellular bacteria, fungi, and viruses [81]. They produce IL-17, IL-22, and IL-23, stimulate many cell types to recruit neutrophils, and promote inflammation at the site of infection [82]. Consequentially, therapeutic strategies directed to neutralizing these cytokines utilizing monoclonal antibodies have shown encouraging results [46,83,84,85,86,87]. By contrast, Tregs express the transcription factor forehead box P3 (Foxp3) and produce anti-inflammatory cytokines like IL-10 and transforming growth factor-β (TGF-β) which inhibit immune responses to control immune homeostasis. These two classes of T cells subsets have opposing roles during inflammatory and immune responses: Th17 can cause, while Treg suppress autoimmune and inflammatory-based diseases [88]. Moreover, Th17 and Treg share a common signaling pathway mediated by TGF-β, but the external milieu present during activation determines these cells’ polarisation [89,90,91]. Considering the “plasticity” of the differentiation process and since in inflammatory states each cell type can convert to the other, it is not unexpected that the equilibrium between Th17/Treg is critical for pathogenesis, prognosis, and therapy of several autoimmune diseases [92,93,94]. 

Indeed, the Th17/Treg ratio is increased in patients with psoriasis, inflammatory bowel disease (IBD), RA [95], and multiple sclerosis (MS) [96].

In Figure 2**,** we have summarized the broad panel of natural compounds which could potentially modulate Th17/Treg function and prove potentially useful in preventing and/or treating different immune-mediated inflammatory diseases [78,93,97].

In vivo pre-clinical studies suggest that green tea and its active ingredient, epi-gallocatechin gallate (EGCG), may effectively improve the symptoms and inflammatory conditions of autoimmune diseases [98,99]. EGCG is also viewed as an anti-inflammatory agent [100] due to its inhibitory effect on the release of IL-6 and IL-17 [101] and the regulation of the Th17/Treg balance [102].

Baicalin, from *Scutellaria baicalensis*, has been shown to effectively reduced inflammation and tissue damage in colitis, and modulate Th17/Treg imbalance in the colon, ameliorating colorectal inflammation [103]. On the other hand, baicalin downregulated the levels of Th17-related cytokines (IL-17 and IL-6), that provoked chronic inhibition of autophagy and induction of claudin-2 expression, leading to apoptosis and dysregulated epithelial barrier function with subsequent gut dysbiosis and intestinal diseases [104]. These findings highlight a novel immunomodulatory role of baicalin not only in bowel inflammation [105] but also in asthma [106,107] and arthritis [108]. 

In addition, glycyrrhizin (or glycyrrhizic acid), from *Glycyrrhiza glabra*, with a corticosteroid-like structure, has been shown to possess several beneficial pharmacological activities, including anti-asthmatic effects [109] and anti-inflammatory activity [110], acting on the Th17/Treg balance and regulating the immune response in different models of chronic inflammation. Noteworthy are mangiferin [111] and neomangiferin [112], main active constituents of *Mangifera indica*, that decreased the proportion of Th17 and the levels of IL-17A, while increasing the balance of Treg and the expression of anti-inflammatory IL-10. 

Th17 and Treg populations are also affected by other polyphenols: curcumin (from *Curcuma longa*) [113] and quercetin (from *Gingko biloba* and *Hypericum perforatum*) attenuatelevels of TNF-α, IL-6 [114] and IL-17 [115], associated with the downregulation of NF-κB [116]. 

Additionally, cryptotanshinone from *Salvia miltiorrhiza* [117], oleanolic acid isolated from *Vigna angularis* [118] and norisoboldine, the main active ingredient of the dry root of *Lindera aggregata* [119], were proven to have anti-arthritic effects, decreasing inflammation and joint destruction. 

Taken together, these findings, in addition to the other medicinal plants and their active components such as berberine (from *Breberis vulgaris*) [120], rosmarinic acid (from *Perilla frutescens*) [121], timosaponin AIII (from *Anemarrhena asphodeloides*) [122], madecassic acid (from *Centella asiatica*) [123], isogarcinol (from *Garcinia mangostana*) [124], astilbin (from Smilax Glabra) [125] and paeoniflorin (from *Paeonia lactiflora*) [126], shown in Figure 2, highlight the important role natural compounds could play in the treatment of the most common autoimmune diseases, through their ability to act on Th17/Treg ratio.

## 5. Conclusions and Future Prospect

European legislation for food supplements and nutraceuticals is not fully harmonized and not adapted to meet current challenges. In this context, the Belgian, French, and Italian authorities have decided to develop a common approach for evaluating botanicals in the BELFRIT project providing harmonization in terms of identification and classification of medicinal plants used in food supplement and/or nutraceuticals. However, in the BELFRIT list, there are only a few and fragmented pieces of information regarding the natural products and compounds targeting COX-2/mPGES-1 coupling enzymes and Th17/Treg repertoire. In this regard, the information reported here (in particular for those secondary plant’s metabolites targeting both pathways, Figure 3) highlights the need for further detailed studies, including mechanism exploration, safety profile, and clinical trials to discover clinically useful drugs and therapeutic targets in widespread pathologies related to inflammatory-based and autoimmune-related diseases.

## Figures and Tables

**Figure 1 molecules-25-06016-f001:**
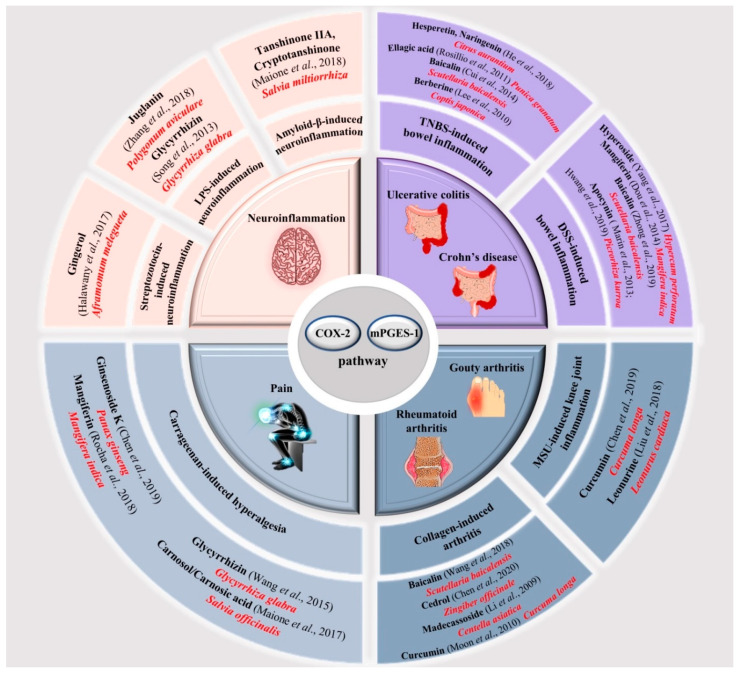
Schematic representation of natural compounds (black bold), by different botanical sources (in red) targeting COX-2/mPGES-1 pathway as alternative therapies for different inflammatory-related diseases. The figure, also, shows the pre-clinical models (black bold), where these active components have been tested with related bibliographic references.

**Figure 2 molecules-25-06016-f002:**
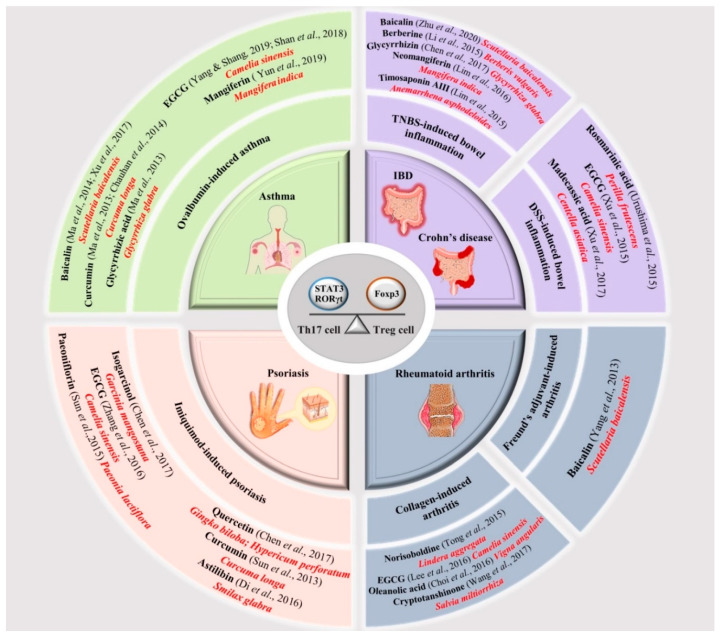
Schematic representation of natural compounds (black bold), by different botanical sources (in red) targeting Th17/Treg ratio as alternative therapies for different autoimmune-based diseases. The figure, also, shows the pre-clinical models (black bold), where these active components have been tested with related bibliographic references.

**Figure 3 molecules-25-06016-f003:**
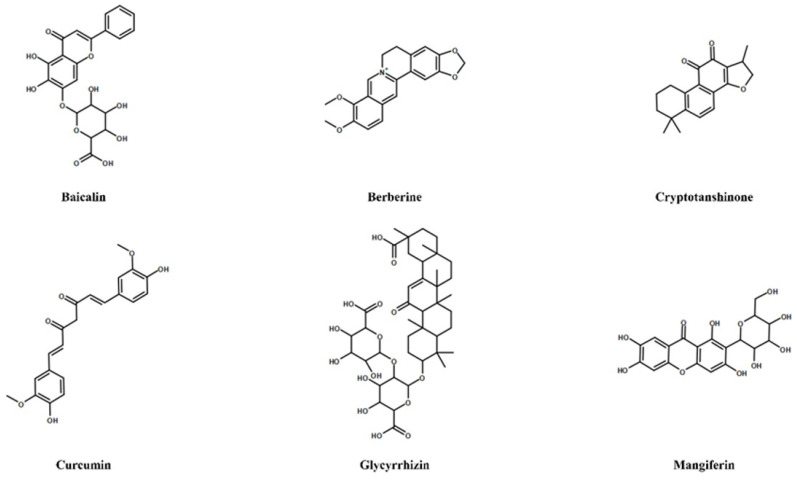
Chemical structures of most significant secondary metabolites targeting both COX-2/mPGES-1 pathway and Th17/Treg axis.

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
