# Peer review of "Present Status and Future Trends of Natural-Derived Compounds Targeting T Helper (Th) 17 and Microsomal Prostaglandin E Synthase-1 (mPGES-1) as Alternative Therapies for Autoimmune and Inflammatory-Based Diseases"

_molecules, 2020, doi:10.3390/molecules25246016_

Round 1
Reviewer 1 Report
The paper is interesting and is well written. Since it is an Opinion Paper, I recommend it's acceptance after minor revision (lines 84-91 are not adequately formatted).
Author Response
- We are grateful to the Reviewer for the time that he/she has dedicated to the improvement of our manuscript. There is a clear commitment to the job that has been undertaken and this could only be praised. Accordingly, we have formatted lines 84-91 (paragraph number 2). All the changes and corrections have been highlighted in the revised version of the manuscript using the "Track Changes" function.
Reviewer 2 Report
This is an opinion paper of the potential anti-inflammatory effects of various natural derived compounds in the treatment of autoimmune and chronic inflammatory disease. Overall, the ms does little to update the status of these compounds, nor does it offer much about future trends in this area. It simply states that this plant or that has been linked to this effect or that in the literature. The opinion that is being presented in no way seems novel or contentious. I think it’s widely accepted that there are a number of natural products that have the potential to be effective therapeutics in the treatment of inflammatory conditions. Perhaps if a more comprehensive overview and/or model of potential mechanism(s) was provided it would be more compelling. There are some typos and grammatical errors in the text.
Author Response
-We are grateful to the Reviewer for the time that he/she has dedicated to the improvement of our manuscript. Together with my co-authors, I have carefully examined your comments and the reports of other independent Reviewers. In this manuscript, we provide a brief overview of natural products/compounds, nowadays included in the Italian list of botanicals to be used in food supplements as well as in the BELFRIT list, on different fields of interests such as inflammation and immunity. In this view, we focus our “opinion” on novel therapeutic target as such as COX-2/mPGEs-1 coupling enzymes and Th17/Treg circulating repertoire. This aspect is completely new in international literature.
-We have presented two Figures with a schematic representation of natural compounds, by different botanical sources targeting COX-2/mPGES-1 pathway (Figure 1) and Th17/Treg ratio (Figure 2) as alternative therapies for different inflammatory-related and autoimmune-based diseases. Both Figures, also, shows the pre-clinical models, where these active components have been tested with related bibliographic references. Our aim was to be novel and original (even in terms of presentation of information’s) but not contentious on the mentioned scientific fields
- We made all the necessary grammatical corrections to the manuscript following careful revision by a native English speaker (Asif J Iqbal), a co-author of this manuscript. All the changes and corrections have been highlighted in the revised version of the manuscript.
Reviewer 3 Report
This manuscript addresses a very important topic, i.e. a limited number of studies and reviews highlight the potential modulation of the coupling enzymatic 26 pathway COX-2/mPGES-1 and Th17/Treg circulating cells. In this manuscript, authors have shown their opinion on novel therapeutic targets such as COX-2/mPGES-1 coupling enzymes and Th17/Treg circulating repertoire. Authors also presents evidence in Figure 1 and Figure 2 that natural products/compounds as previously reported could interfere with COX-2/mPGES-1 signalling pathway. Therefore, natural compounds can provide an alternative treatment of inflammatory and autoimmune diseases.
Although the structure of this review is clear and provides update of knowledge to readers, I still would like to make several Specific Questions/Comments as follows:
1. Abstract section: Line 29, authors may consider to use opinion to replace “opinion”.
2. Introduction Section: In the first paragraph, Line 47 "now guarantees the safety, quality, and effectiveness of food". Please change effectiveness to health promoting prosperities, because the effectiveness of food supplements is yet to be clinically validated.
3. Introduction Section: Lines 51-53, "and quite often apply modern standards of effectiveness testing to herbs and medicines derived from natural sources, perform high-quality clinical trials and standards for purity or dosage exist". Please amend it as "and quite often it applies modern standards of effectiveness testing to herbs and medicines derived from natural sources, performs high-quality clinical trials and uses standards for purity or dosage".
4. Introduction Section: Lines 53-54, "In this scenario, we must consider that many botanicals are used in both food supplements, and nutraceuticals and a precise, unique, and standardized definition/s and". Please change it to "In this scenario, many botanicals are used in both food supplements and nutraceuticals, and yet a precise, unique, and standardized definition/s and".
5. Why did authors present the Figure1 and Figure 2 at 90 degree rotation? It will be more eye catching if the Figures are set at 0 degree rotation.
6. This is the last comments, but important: It has been widely recognised that anti-inflammatory natural products with multiple targets. However, two highly selective COX-2 inhibitors, Celecoxib and Rofecoxib, were marked, but halted in 2004 when Rofecoxib was withdrawn due to serious cardiovascular events. Have authors considered sides effects of natural products/compounds which may need to address? The example is glycyrrhizin from Glycyrrhiza glabra (Page 4, Line 146), which is safe, but may cause hypertension or pulmonary edema in some people.
Author Response
- We are grateful to the Reviewer for the time that he/she has dedicated to the improvement of our manuscript and for the evaluation. There is a clear commitment to the job that has been undertaken and this could only be praised. Together with my co-authors, I have carefully examined your comments and we hope that our proposed replies and revisions address the concerns raised and improve the overall quality of our manuscript. All the changes and corrections have been highlighted in the revised version of the manuscript using the "Track Changes" function.
- Abstract section: Line 29, authors may consider to use opinion to replace “opinion”.
- We have replaced “opinion” with opinion.
- Introduction Section: In the first paragraph, Line 47 "now guarantees the safety, quality, and effectiveness of food". Please change effectiveness to health promoting prosperities, because the effectiveness of food supplements is yet to be clinically validated.
- We agree with Reviewer regarding this aspect. As suggested, we have changed and updated this sentence.
- Introduction Section: Lines 51-53, "and quite often apply modern standards of effectiveness testing to herbs and medicines derived from natural sources, perform high-quality clinical trials and standards for purity or dosage exist". Please amend it as "and quite often it applies modern standards of effectiveness testing to herbs and medicines derived from natural sources, performs high-quality clinical trials and uses standards for purity or dosage".
- We agree with Reviewer regarding this aspect. As suggested, we have changed and updated this sentence.
- Introduction Section: Lines 53-54, "In this scenario, we must consider that many botanicals are used in both food supplements, and nutraceuticals and a precise, unique, and standardized definition/s and". Please change it to "In this scenario, many botanicals are used in both food supplements and nutraceuticals, and yet a precise, unique, and standardized definition/s and".
- We agree with Reviewer regarding this aspect. As suggested, we have changed and updated this sentence.
- Why did authors present the Figure1 and Figure 2 at 90 degree rotation? It will be more eye catching if the Figures are set at 0 degree rotation.
- We Thank the Reviewer for this comments. We have preferred to present the Figure 1 and Figure 2 at 90 degree rotation (horizontal format) in order to improve their size and consequently facilitate the reader's understanding.
- This is the last comments, but important: It has been widely recognised that anti-inflammatory natural products with multiple targets. However, two highly selective COX-2 inhibitors, Celecoxib and Rofecoxib, were marked, but halted in 2004 when Rofecoxib was withdrawn due to serious cardiovascular events. Have authors considered sides effects of natural products/compounds which may need to address? The example is glycyrrhizin from Glycyrrhiza glabra (Page 4, Line 146), which is safe, but may cause hypertension or pulmonary edema in some people.
- We Thank the Reviewer for this important comment. In this opinion article. We have not considered to include the “side effects” of of natural products/compounds. As reported inj Figure 1 and 2, we have provided a schematic representation of natural compounds, by different botanical sources targeting COX-2/mPGES-1 pathway and Th17/Treg ratio as alternative therapies for different inflammatory-related and autoimmune-based diseases. As stated in the paragraph 2, we have identified only original articles in English that evaluated pre-clinical studies, in vivo rodents‘ models, and isolated and/or well-characterized compounds/extracts from all articles.
Round 2
Reviewer 2 Report
The revised ms is not appreciably different from the original. The authors response does not really address my concerns; simply adding a few sentences is not a significant upgrade.
Author Response
R: The revised ms is not appreciably different from the original. The authors response does not really address my concerns; simply adding a few sentences is not a significant upgrade.
A: We thank the Reviewer for the time that he/she has dedicated to the improvement of our manuscript. We apologize for the misunderstanding generated by our first response. We will try to be more detailed and clear.
A: As previously reported in the first set of revision, in this manuscript, we provide a brief overview of natural products/compounds, nowadays included in the Italian list of botanicals to be used in food supplements as well as in the BELFRIT list, on different fields of interests such as inflammation and immunity. In this view, we focus our “opinion” on novel therapeutic target as such as COX-2/mPGEs-1 coupling enzymes and Th17/Treg circulating repertoire. This aspect is completely new in international literature. For the reaosns reported, we think that we have addressed the concern/s raised by this Reviewer (R: “….the ms does little to update the status of these compounds, nor does it offer much about future trends in this area. It simply states that this plant or that has been linked to this effect or that in the literature).
A: We have presented two original Figures with a schematic representation of natural compounds, by different botanical sources targeting COX-2/mPGES-1 pathway (Figure 1) and Th17/Treg ratio (Figure 2) as alternative therapies for different inflammatory-related and autoimmune-based diseases. Both Figures, also, shows the pre-clinical models, where these active components have been tested with related potential mechanism(s) and bibliographic references. Our aim was to be novel and original (even in terms of presentation of information’s) but not contentious on the mentioned scientific fields. For the reaosns reported, we think that we have addressed the concern/s raised by this Reviewer (R: “….The opinion that is being presented in no way seems novel or contentious. I think it’s widely accepted that there are a number of natural products that have the potential to be effective therapeutics in the treatment of inflammatory conditions. Perhaps if a more comprehensive overview and/or model of potential mechanism(s) was provided it would be more compelling…”).
R: ”…There are some typos and grammatical errors in the text”. “…The authors response does not really address my concerns; simply adding a few sentences is not a significant upgrade…”.
A: We made all the necessary grammatical corrections to the manuscript following careful revision by a native English speaker (Asif J Iqbal), a co-author of this manuscript. All the changes and corrections have been highlighted in the revised version of the manuscript. We would appreciate if Reviewers could indicate line-by-line typos and grammatical errors that are still present in the revised version of the manuscript.